# Objectively and subjectively measured physical activity levels in individuals with whiplash associated disorder and aged-matched healthy controls

Carrie Ritchie[1], Esther Smits[1], Nigel Armfield[1,2], Michele Sterling[1]*

1 RECOVER Injury Research Centre and National Health and Medical Research Council (NHMRC) Centre for Research Excellence: Better Health Outcomes After Compensable Injury, University of Queensland, Brisbane, Queensland, Australia, 2 Centre for Health Services Research, Faculty of Medicine, The University of Queensland, Brisbane, Queensland, Australia

* m.sterling@uq.edu.au

**Data Availability Statement:** All relevant data are within the paper and its Supporting information files.

## Abstract

### Background

Whiplash associated disorders (WAD) are the most common non-hospitalised injuries resulting from a motor vehicle crash. Half of individuals with WAD experience ongoing pain and disability. Furthermore, individuals with persistent WAD have lower levels of aerobic capacity and isometric strength compared with age-matched controls. It is not known whether these differences are associated with increased levels of pain and disability, or with reduced physical activity (PA) participation.

### Objective

Our primary aim was to compare PA levels in individuals with persistent WAD with healthy controls. Secondary aims were to: compare objective and subjective measurements of PA; explore factors that may influence PA; and describe proportions of these populations meeting World Health Organisation PA guidelines.

### Methods

Objective (ActiGraph accelerometer; seven days) and subjective (International Physical Activity Questionnaire (IPAQ)) PA data were collected for n = 53 age-matched participants (WAD n = 28; controls n = 25).

### Results

Independent sample t-tests showed no significant difference in objectively measured PA (p>0.05) between WAD and controls. For the subjective measure (IPAQ), controls reported more overall weekly PA (t = 0.219, p<0.05), while WAD participants reported more weekly walking minutes (t = -0.712, p<0.05). Linear regression showed mental health quality-of-life predicted objectively measured moderate intensity PA ($R^2$ = 0.225, F (2, 44) = 6.379,

**Funding:** The authors received no specific funding for this work.

**Competing interests:** The authors have declared that no competing interests exist.

p<0.004) and subjectively reported overall PA ($R^2 = 0.132$, F (1, 41) = 6.226, p<0.017). Bland-Altman analyses indicated that subjects over-reported MVPA and under-reported sedentary time using the IPAQ.

## Conclusions

Individuals with WAD had levels of physical and mental health quality-of-life significantly lower than controls and below population norms yet participated in similar levels of PA. Given that increased perceptions of mental health quality-of-life were positively associated with objectively measured MVPA and subjectively reported overall PA, strategies to help people with WAD achieve adequate doses of MVPA may be beneficial. ActiGraph-measured and IPAQ-reported PA were discordant. Hence, IPAQ may not be a reliable measure of habitual PA in WAD.

## Introduction

Whiplash associated disorders (WAD) is the term used to describe a cluster of symptoms, including neck pain and disability, that typically result from an acceleration/deceleration movement of the neck following a motor vehicle crash (MVC). WAD are the most common non-hospitalised injuries resulting from a MVC, estimated to account for approximately 75% of all survivable MVC injuries in Australia [1], and over 95% in the USA [2]. Over the past few decades, recovery rates have remained unchanged with approximately 50% of individuals experiencing on-going pain, disability and psychological distress [3, 4]. Additionally, individuals with persistent WAD have lower levels of aerobic capacity and isometric strength compared with age-matched healthy controls [5, 6]. It is not known if this reduced physical fitness is associated with increased levels of pain and disability, or with reduced physical activity (PA) participation.

Most intervention trials in WAD have focussed on therapeutic exercise and have not addressed the need for promoting sustained PA participation in WAD [7]. Thus, clinical practice guidelines for treating WAD focus largely on neck specific exercises including range of motion and low load isometric exercises to optimise function and prevent disability [8–10]. Information about the inclusion of habitual physical activity to improve physical and psychological health is lacking.

While the benefits of participation in regular PA are numerous and accepted [11], the relationship between participation in regular PA and musculoskeletal pain is less clear. In individuals with chronic non-traumatic neck pain, reduced PA is associated with an increased risk of ongoing neck pain [12]. Some data suggest that individuals with chronic non-traumatic neck pain may be less physically active than age-matched healthy controls [12, 13], whereas other data suggest that PA levels are similar [14, 15]. Given that individuals with WAD have higher neck disability [16, 17] and pain [17], as well as lower health-related quality of life (HRQoL) [16, 18] compared with individuals with non-traumatic neck pain, it may be that individuals with persistent WAD are insufficiently active for good health, thereby, increasing their risk of preventable morbidity and mortality, and compounding the effects of the WAD.

Current World Health Organisation (WHO) guidelines recommend that adults should accumulate 150 to 300 minutes of moderate intensity PA or 75 to 150 minutes of vigorous intensity PA, or an equivalent combination of both, each week [11]. It is not known whether individuals with WAD meet the WHO recommendations. Habitual PA can be measured using

objective (e.g., accelerometry) or subjective (e.g., self-report) measures. Objective tools provide increased reliability and accuracy since they do not rely on recall and thereby avoid biases associated with self-reporting [19]. However, accelerometers can be expensive and data analyses oftentimes require research expertise [19]. Self-report tools that have shown acceptable measurement properties and may be a more feasible way to assess habitual PA in adults [20]. The International Physical Activity Questionnaire (IPAQ) is the most widely used self-report PA questionnaire with the long version (IPAQ-L) shown to be valid for both research purposes and clinical practice [20, 21].

Our primary aim was to compare PA levels in individuals with persistent WAD and aged-matched healthy controls. The secondary aims were to compare objective (ActiGraph accelerometer) and subjective (IPAQ-L) measurements of PA, to describe the proportions of participants meeting WHO guidelines for PA, and to explore factors that may influence PA.

## Materials and methods

### Participants

Participants were individuals with persistent WAD and aged-matched healthy controls.

Inclusion criteria were: aged 18–65 years; ambulatory; and able to wear an accelerometer for seven days. Additional inclusion criteria for individuals with WAD: Quebec task force WAD grade II or III of at least three months duration [22] and reported presence of neck pain on a numeric rating scale of $\geq$ 3/10 over the past week [23]. Participants with WAD were excluded if they had: a serious spinal pathology; undergone spinal surgery in the past 12 months; a confirmed fracture or dislocation at time of injury (e.g., WAD Grade IV); or nerve root compromise. Additional exclusion criteria for healthy controls were: a previous whiplash injury, or history of neck pain.

### Procedure

Participants were invited to attend two sessions in Brisbane, Queensland, Australia. At session one, participants completed a baseline questionnaire and were provided with the ActiGraph GT9X Link (ActiGraph LLC, Pensacola, FL, USA) accelerometer. Participants were asked to continue with their usual routine for seven days. Seven to 10 days after the first session, participants returned the accelerometer and completed the IPAQ-L [21].

The baseline questionnaire included demographic data (age, body height, body weight, working status, and compensation status), and the following validated tools:

- the neck disability index (NDI) (0–100%) to assess neck pain and disability [24] with scores $\geq$ 30% indicating moderate to severe neck disability [25];

- the Medical Outcomes Study Short Form Health Survey (SF12) to assess perceived quality of life related to physical functioning and mental health [26] with composite scores < 50 indicating lower than normal quality of life [27]; and

- the pain self-efficacy questionnaire to assess confidence in performing tasks despite pain [28] with scores < 20 indicating low pain self-efficacy [29].

### Objective measure of PA

Participants were asked to wear the ActiGraph GT9X Link, a commonly used research-grade accelerometer, on their non-dominant wrist continually for seven days. Data were processed using ActiLife software (v6.13.3) with the devices initialised to record accelerations at 30 Hz

and vector magnitude counts integrated over 60 second epochs. Data were exported to MATLAB R2017b (The MathWorks, Inc., 2017, Natick, MA) for analysis of non-wear time, valid wear time and conversion to intensity of PA. Non-wear time was defined as $\geq 90$ consecutive minutes of 0 vector magnitude counts, with an allowance for up to two minutes of non-zero counts to allow for detection of movement artifacts. Aligned with recommendations, a valid wear day was defined as $\geq 10$ hours of wear time, and a valid week was $\geq 4$ valid wear days [30]. The outcome for total habitual PA was ActiGraph counts per day. Additional outcomes were minutes per day (mpd) spent sedentary, and in light, moderate, vigorous and moderate/vigorous intensity PA (MVPA). Counts were converted to PA intensity levels using wrist-developed cut-points for ActiGraph accelerometers (sedentary $\leq 2860$ cpm, light = 2861–4835 cpm, moderate = 4836–8452 cpm, vigorous = $\geq 8453$ cpm) [31, 32].

## Subjective measure of PA

The IPAQ-L is a validated, self-report survey used to subjectively measure habitual PA [20, 21]. The amount of time spent walking and participating in moderate or vigorous PA is reported for the previous seven days across various domains (e.g., transport, work, home, and leisure time). Time spent sitting is also estimated. In the present study, participants were asked to respond to the questions for the seven-day period in which they wore the ActiGraph. To optimise the accuracy of participant's reflection of their PA, only IPAQ-L data completed within three days of the ActiGraph wear-week were included in the analyses. The IPAQ scoring manual (www.ipaq.ki.se) provided the information to compute volume of PA by weighting each type and intensity of PA by its energy requirement defined in metabolic equivalents (MET). For example, while one MET represents the amount of energy expended while sitting quietly, moderate intensity PA was defined as four METs. MET-minutes were calculated by multiplying the MET score of an activity by the number of activity minutes. The outcome measure for total PA was total MET–minutes of PA per week, calculated by summing total walking, moderate, and vigorous MET-minutes. Additional outcomes were mpd spent sitting, walking, and participating in MVPA.

## Sample size

Since there are no published PA data for individuals with persistent WAD, data from a population study of ActiGraph measured PA in adults was used to calculate sample size [33]. Based on the difference of predicted means of 25% and standard deviation equal to 30% of the mean, with significance at $p < 0.05$ and power of 80% ($\beta = 0.20$), a sample size of 23 participants was needed in each group. A sample size of 46 participants was deemed sufficient to test the prediction of objectively or subjectively measured PA using four baseline measures based on the assumption that 10 cases are needed per predictor variable.

## Data analysis

Baseline measures (e.g., sex, age, body mass index (BMI), SF12PCS, SF12MCS), and objective/subjective measures of PA and sedentary time were tested for normality using the Kolmogorov-Smirnov test. Comparisons were conducted using t-tests for normally distributed data, or Mann Whitney U otherwise. The proportion of WAD/control participants meeting WHO PA guidelines [11] was assessed by chi-squared analysis. To test whether baseline measures were associated with objective and subjective measures of PA, we used univariate linear regression followed by multivariate linear regression with those variables with a relationship of $p < 0.1$ from the corresponding univariate analysis.

The relationship between the objective and subjective measures of total PA, MVPA and sedentary time were assessed using Pearson-product moment correlation. In addition, the Bland-Altman method was used to describe agreement between objectively and subjectively measured MVPA and sedentary time with 95% limits of agreement used to explain total error [34]. The difference between the objective and subjective methods was assessed with a paired t test. SPSS (version 28.0) was used for all statistical analyses. Cohen's criteria (r = 0.1, 0.3, 0.5 small, medium and large respectively) [35] were used to interpret effect size.

### Ethics

Ethical approval was granted by the University of Queensland (#2019000085). The study was conducted in accordance with Declaration of Helsinki, and all participants provided written informed consent prior to participating in the study.

## Results

We recruited 28 individuals with WAD and 25 controls. Most were female (WAD n = 23 (82%); controls n = 21 (84%)). Most participants were currently employed (WAD n = 23 (82%); controls n = 25 (100%)), though 11 (39%) WAD participants were not working or working reduced hours because of neck pain. Twenty-four (86%) WAD participants had submitted a third-party compensation claim. WAD participants had moderate levels of neck disability [25] and mildly reduced levels of confidence to participate in activities while in pain [29] (Table 1). There were no significant differences between the groups in age or BMI (Table 1). However, compared with controls, WAD had significantly lower mental (U = 127.0, p<0.000) and physical (U = 2.0, p<0.000) HRQoL (Table 1). There were insufficient males in the sample to allow any assessment by sex.

### Objectively measured PA

A valid wear week was completed by 82% (n = 23) of WAD and 96% (n = 24) of control participants. ActiGraph wear time was not significantly different between the groups (Table 2). There were no significant differences between WAD and control in objectively measured total PA, or mpd of light, moderate, or vigorous intensity PA, MVPA, or sedentary time (Table 2). There was no significant difference between the number of WAD (n = 16 (70%)) and controls (n = 19 (79%)) meeting the WHO guidelines for PA ($\chi2$ = 0.569, df = 1, p<0.517) [11].

**Table 1. The median [interquartile range], and results of Mann Whitney U for baseline measures.**

|  | WAD (n = 28) | Controls (n = 25) | U, z |
|---|---|---|---|
| NDI (%) | 40 [12, 58] |  |  |
| PSEQ | 30 [4, 56] |  |  |
| Age | 43 [35, 52] | 45 [35, 54] | 339.5, -0.187 |
| BMI | 26.7 [21.7, 30.5] | 23.8 [22.2, 26.0] | 417.5, 1.496 |
| SF12MCS | 44.3 [35.5, 51.3] | 57.8 [55.0, 57.9] | 127.0, -3.857** |
| SF12PCS | 37.5 [33.3, 44.2] | 56.6 [55.5, 57.0] | 2.0, -6.148** |

BMI (body mass index), NDI (neck disability index), PSEQ (pain self-efficacy questionnaire), SF12MCS (the medical outcomes short form mental component score), SF12PCS (the medical outcomes short form physical component score), WAD (whiplash associated disorder)

** p<0.000

**Table 2. Mean (standard deviation), and results of independent samples t tests for ActiGraph and IPAQ measures.**

| ActiGraph | WAD (n = 23) | Controls (n = 24) | t |
|---|---|---|---|
| Wear time mpd | 921.5 (79.3) | 916.0 (92.3) | -0.218 |
| PA counts per day | 238.2 x 10$^4$ (63.2 x 10$^4$) | 256.6 x 10$^4$ (70.3 x 10$^4$) | 0.946 |
| Light intensity mpd | 337.2 (71.4) | 332.6 (84.1) | -0.202 |
| Moderate intensity mpd | 16.5 (8.2) | 17.2 (7.6) | 0.322 |
| Vigorous intensity mpd | 27.5 (15.3) | 35.4 (19.5) | 1.556 |
| MVPA mpd | 43.9 (22.8) | 52.6 (25.3) | 0.985 |
| Sedentary mpd | 540.3 (102.4) | 530.7 (108.4) | -0.312 |
| **IPAQ-L** | WAD (n = 20) | Controls (n = 24) | |
| MET-minutes per week | 2906.2 (2302.2) | 3030.4 (1431.0) | 0.219* |
| Walking mpd | 44.7 (45.7) | 36.5 (26.3) | -0.712* |
| Moderate intensity mpd | 51.1 (49.7) | 51.8 (37.0) | -0.053 |
| Vigorous intensity mpd | 9.4 (17.2) | 16.3 (19.5) | 1.220 |
| MVPA mpd | 60.5 (61.4) | 68.0 (48.6) | 0.454 |
| Sedentary mpd | 379.7 (182.5) | 378.6 (152.0) | -0.022 |

mpd (minute per day), MVPA (moderate-vigorous intensity physical activity), PA (physical activity), WAD (whiplash associated disorder)

*p < 0.05

The results of univariate linear regressions are shown in Table 3. While SF12PCS was a predictor of ActiGraph measured PA counts per day, this variable explained only 9% of the variance ($R^2$ = 0.090, F (1, 45) = 4.453, p<0.040). A standard multiple regression showed that MVPA mpd was predicted by age and SF12MCS ($R^2$ = 0.225, F (2, 44) = 6.379, p<0.004) where SF12MCS was the only variable to contribute significantly to the model (B = 0.972, p<0.004).

## Subjectively measured PA

The IPAQ-L questionnaire was completed between days 7 and 10 post-baseline by 71% (n = 20) of WAD and 96% (n = 24) of control participants. Controls reported to participate in

**Table 3. Univariate linear regression analyses assessing the relationship between objective and subjective PA measures and baseline variables.**

| | Age | | | BMI | | | SF12PCS | | | SF12MCS | | |
|---|---|---|---|---|---|---|---|---|---|---|---|---|
| | B | t | p | β | t | p | β | t | p | β | t | p |
| ActiGraph | | | | | | | | | | | | |
| PA counts per day | -2854.86 | -0.307 | 0.761 | -23485.99 | -1.422 | 0.162 | 19175.37 | 2.110 | 0.040* | 12547.09 | 1.284 | 0.206 |
| MVPA mpd | -0.573 | -0.252 | 0.087* | -0.455 | -0.747 | 0.459 | 0.515 | 1.526 | 0.134 | 0.981 | 2.973 | 0.005* |
| Sedentary mpd | 0.177 | 0.121 | 0.904 | 3.926 | 1.526 | 0.134 | -1.372 | -0.930 | 0.357 | -2.331 | -1.538 | 0.131 |
| IPAQ-L | | | | | | | | | | | | |
| MET-minutes per week | 2.193 | 0.082 | 0.935 | -77.363 | -1.645 | 0.108 | 38.950 | 1.441 | 0.157 | 67.245 | 2.495 | 0.017* |
| MVPA mpd | -0.168 | -0.214 | 0.831 | -1.418 | -1.011 | 0.318 | 1.116 | 1.411 | 0.166 | 1.220 | 1.481 | 0.146 |
| Sedentary mpd | -7.290 | -3.454 | 0.001* | -0.147 | -0.034 | 0.973 | 0.707 | 0.285 | 0.777 | -0.133 | -0.051 | 0.959 |

BMI (body mass index), mpd (minute per day), MVPA (moderate-vigorous intensity physical activity), PA (physical activity), SF12MCS (the medical outcomes short form mental component score), SF12PCS (the medical outcomes short form physical component score), WAD (whiplash associated disorder)

*P<0.1 and therefore eligible to be included in multivariate analyses

significantly more overall PA per week than WAD participants (t = 0.219, p<0.024) (Table 2). While there were no significant differences between the groups in terms of reported MVPA, WAD participants reported significantly more walking mpd than controls (t = -0.712, p<0.025) (Table 2). There was no significant difference between the number of WAD participants (n = 13 (65%)) and controls (n = 19 (79%)) meeting the WHO guidelines for PA ($\chi$2 = 1.104, p<0.329) [11].

The results of univariate linear regressions showed that SF12MCS was a predictor of IPAQ-L total weekly MET-minutes, explaining 13% of the variance ($R^2$ = 0.132, F (1, 41) = 6.226, p<0.017) (Table 3). Furthermore, age was a predictor of reported daily sedentary minutes, explaining 22% of the variance ($R^2$ = 0.225, F (1, 41) = 11.928, p<0.001).

## Relationship of objective and subjective PA measures

Pearson-product moment correlational analysis showed a moderate correlation between the objective (Actigraph PA counts/day) and subjective (IPAQ-L MET-minutes/week) measures of overall PA (r = 0.477, p<0.002) (Fig 1).

A small correlation was found between objectively and subjectively measured MVPA (r = 0.362, p<0.020), though Bland-Altman analysis highlighted the over-reporting (IPAQ) of MVPA by a mean of 17 mpd and showed wide limits of agreement (-119.7 and 85.7 mpd) (Fig 2). Paired t- test showed a significant difference between the measures (ActiGraph mean = 48.4 mpd; IPAQ-L mean = 65.4 mpd; t = -2.077, p<0.022).

While a small correlation was also found between objective and subjective measures of sedentary time (r = 0.371, p<0.019), Bland-Altman analysis highlighted the under-reporting (IPAQ) of sedentary time by a mean of 147 mpd and showed wide limits of agreement (-171 and 464 mpd) (Fig 3). Paired t- test showed a significant difference between the measures (ActiGraph mean = 532.2 mpd; IPAQ-L mean = 385.4 mpd; t = 5.722, p<0.000).

## Discussion

Objectively measured habitual PA, MVPA and sedentary time were similar for WAD and controls. On the other hand, there were some differences in the subjective reporting of weekly PA: controls reported significantly more overall weekly PA, and participants with WAD reported significantly more walking minutes. For WAD participants, levels of physical and mental

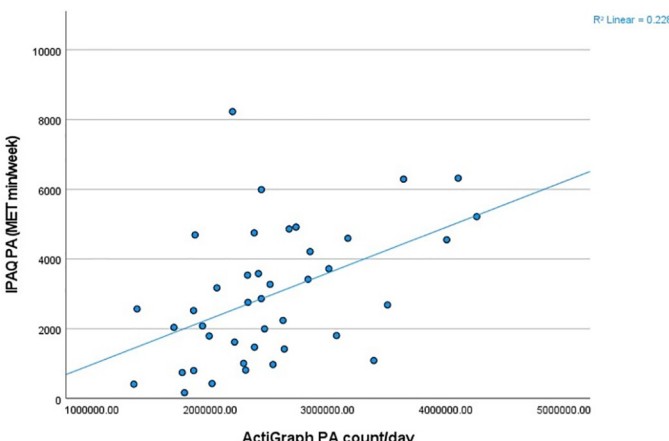

**Fig 1. Scatterplot of total PA measured objectively (ActiGraph total PA counts/day) and subjectively (IPAQ-L reported MET-minutes/week).**

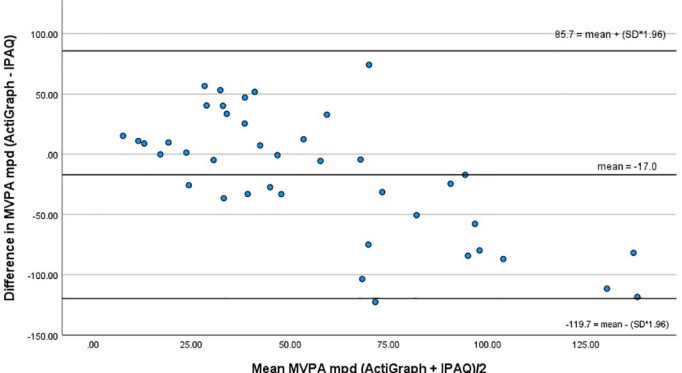

**Fig 2. Bland-Altman plot for MVPA.**

health-related quality of life were below population norms, and significantly lower than controls with increased perceptions of mental health quality of life positively associated with objectively measured MVPA and subjectively reported overall PA. Given the very low physical and mental health quality of life perceived by WAD participants and the well accepted benefits of participation in MVPA for health and quality of life, it may be beneficial to include strategies within clinical guidelines to help people with WAD achieve adequate doses of MVPA.

Participation in any level of physical activity is better than no participation though the health benefits are optimal if weekly PA includes 150–300 minutes of moderate intensity PA or 75 to 150 minutes of vigorous intensity PA, or an equivalent combination of both [11]. We found no significant difference in the percentage of WAD or controls who met these WHO guidelines. Moreover, participants appeared to engage in more habitual MVPA than the general Australian population. Seventy percent of WAD and 79% of HC met WHO guidelines for MVPA. Whereas, recent Australian data showed only 59% of Australian adults completed 30 minutes of aerobic PA on five or more days per week (https://www.abs.gov.au/statistics/health/health-conditions-and-risks/physical-activity/latest-release, accessed 22 August, 2022). Wearing a monitor may have motivated participants to increase their activity, though to minimise extrinsic motivation, the monitor only displayed time and date. While persistent neck disability from a whiplash injury did not appear to inhibit PA participation for many participants, 30% of WAD participants were not meeting current PA guidelines.

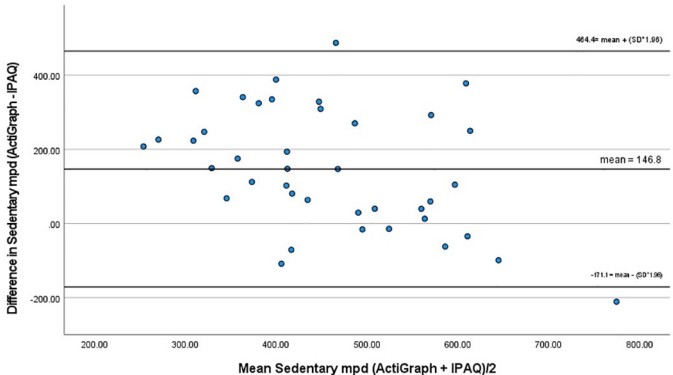

**Fig 3. Bland-Altman plot for sedentary time.**

Both median physical and mental HRQoL for WAD were below population norms, levels consistently shown in WAD [18, 36]. While physical HRQoL had a small positive association with objectively measured overall PA, increased mental HRQoL predicted increased objectively measured MVPA. Previous research has shown physical, but not mental, HRQoL to be significantly associated with objective and self-reported MVPA in older Europeans (>65 years) [37] and self-reported weekly PA in individuals with rheumatic and musculoskeletal diseases [38]. It may be that high symptom levels of depression, anxiety, and posttraumatic stress disorder reported in WAD [39] affect not only mental health quality of life but also participation in MVPA. Participation in PA improves perceptions of quality of life. Previous research by our group showed SF12PCS and SF12MCS scores increased to levels above population norms in participants with persistent WAD following participation in a 12-week PA intervention [36]. Similarly, in general populations, both physical and mental HRQoL scores have been shown to improve following increased frequency of moderate PA [40], with a positive dose-response relationship between weekly MVPA and HRQoL reported across age groups [41]. While therapeutic exercise is an important component of clinical guidelines and research with WAD, including information about the importance of habitual PA and strategies to help achieve adequate doses of MVPA would be valuable for this population.

Based on our results, it is difficult to recommend the IPAQ-L as a self-report measure to assess habitual PA in WAD. Small to moderate correlations were found between objective and subjective PA data though further analysis of MVPA and sedentary time indicated that neither were concordant. Participants over-reported MVPA and under-reported sedentary time with older participants reporting significantly less sedentary time. While WAD participants reported significantly more walking time and less overall PA compared with controls, no significant differences were found for overall objectively measured habitual PA between WAD and controls. No statistical comparisons of objective versus subjective light activity were performed since the IPAQ-L uses walking as a proxy for light intensity PA. A similar lack of agreement has been reported in systematic reviews with weak correlations between subjective self-report measures and accelerometry [19, 42]. Although self-report measures are inexpensive and convenient and the IPAQ-L has been shown to be valid and reliable [20, 21], subjective measures rely on an individual's ability to estimate PA intensity and recall all types of activity. On the other hand, accelerometers such as the ActiGraph provide less overall measurement error and more acute estimates of PA, yet data processing is complex with outcomes such as accelerometer counts requiring valid conversions to meaningful measures such as PA intensity [19, 42]. While accelerometer devices such as the ActiGraph may not be feasible for all research or clinical purposes, the growth of PA measurement devices and wearables provides alternatives that may be more cost effective and require less expertise in analysing the data.

## Study limitations

This study provides much needed data about PA levels in individuals with WAD. Nevertheless, limitations exist. Firstly, neither the IPAQ nor ActiGraph accurately collect information about participation in muscular strength activities, an important component of WHO PA recommendations [11]. Further research is needed to determine if people with WAD achieve recommended doses of muscular strength activities. Secondly, compared with hip-worn ActiGraph devices, wrist-worn ActiGraph devices have better wear compliance but may overestimate MVPA [43]. However, to minimise potential overestimations, wrist-developed cut-points were used to convert ActiGraph proprietary counts to PA intensities [31]. Finally, the study was powered to assess differences in ActiGraph measured PA. It is not known if these numbers are adequate to detect PA differences using the IPAQ-L.

## Conclusions

This was the first study to objectively and subjectively measure habitual PA in WAD. Individuals with a persistent moderate disability from a whiplash injury had levels of physical and mental health-related quality of life significantly lower than controls and below population norms yet participated in similar levels of PA. Given that increased perceptions of mental health quality of life were positively associated with objectively measured MVPA and subjectively reported overall PA, it may be beneficial to include strategies within clinical guidelines to help people with WAD achieve adequate doses of MVPA. Further research is needed to identify the most appropriate methods to assess PA in WAD given the lack of concordance in PA measured with the ActiGraph and reported through the IPAQ.

## Supporting information

**S1 Appendix. Baseline, accelerometer and IPAQ data.**
(XLSX)

## Acknowledgments

The authors acknowledge and thank the Master of Physiotherapy students at the University of Queensland for assisting with data collection.

## Author Contributions

**Conceptualization:** Carrie Ritchie, Esther Smits, Michele Sterling.

**Data curation:** Carrie Ritchie, Esther Smits, Nigel Armfield, Michele Sterling.

**Formal analysis:** Carrie Ritchie, Esther Smits, Nigel Armfield.

**Investigation:** Carrie Ritchie, Esther Smits.

**Methodology:** Carrie Ritchie, Esther Smits, Nigel Armfield, Michele Sterling.

**Project administration:** Carrie Ritchie.

**Resources:** Carrie Ritchie, Esther Smits.

**Software:** Esther Smits, Nigel Armfield.

**Supervision:** Carrie Ritchie, Esther Smits, Michele Sterling.

**Validation:** Carrie Ritchie, Esther Smits, Nigel Armfield, Michele Sterling.

**Visualization:** Carrie Ritchie, Esther Smits, Nigel Armfield, Michele Sterling.

**Writing – original draft:** Carrie Ritchie, Esther Smits, Nigel Armfield, Michele Sterling.

**Writing – review & editing:** Carrie Ritchie, Esther Smits, Nigel Armfield, Michele Sterling.

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
