## [Decision Letter · Decision Letter 0]

4 Aug 2023

PONE-D-23-11007Objectively and subjectively measured physical activity levels in individuals with whiplash associated disorder and aged-matched healthy controlsPLOS ONE

Dear Dr. Ritchie,

Thank you for submitting your manuscript to PLOS ONE. After careful consideration, we feel that it has merit but does not fully meet PLOS ONE’s publication criteria as it currently stands. Therefore, we invite you to submit a revised version of the manuscript that addresses the points raised during the review process.

ACADEMIC EDITOR:Dear Authors,one expert in the field reviewed your manuscript finding several minor issues you should consider during the revision process.============================== Please submit your revised manuscript by Sep 18 2023 11:59PM. If you will need more time than this to complete your revisions, please reply to this message or contact the journal office at plosone@plos.org. Please include the following items when submitting your revised manuscript:A rebuttal letter that responds to each point raised by the academic editor and reviewer(s). You should upload this letter as a separate file labeled 'Response to Reviewers'.A marked-up copy of your manuscript that highlights changes made to the original version. You should upload this as a separate file labeled 'Revised Manuscript with Track Changes'.An unmarked version of your revised paper without tracked changes. You should upload this as a separate file labeled 'Manuscript'.If applicable, we recommend that you deposit your laboratory protocols in protocols.io to enhance the reproducibility of your results. Protocols.io assigns your protocol its own identifier (DOI) so that it can be cited independently in the future. For instructions see: https://journals.plos.org/plosone/s/submission-guidelines#loc-laboratory-protocols. Additionally, PLOS ONE offers an option for publishing peer-reviewed Lab Protocol articles, which describe protocols hosted on protocols.io. Read more information on sharing protocols at https://plos.org/protocols?utm_medium=editorial-email&utm_source=authorletters&utm_campaign=protocols.

We look forward to receiving your revised manuscript.

Kind regards,

Emiliano Cè

Academic Editor

PLOS ONE

Journal Requirements:

Reviewers' comments:

Reviewer's Responses to Questions

**Comments to the Author**

1. Is the manuscript technically sound, and do the data support the conclusions?

Reviewer #1: Yes

2. Has the statistical analysis been performed appropriately and rigorously? 

Reviewer #1: Yes

3. Have the authors made all data underlying the findings in their manuscript fully available?

Reviewer #1: Yes

4. Is the manuscript presented in an intelligible fashion and written in standard English?

Reviewer #1: Yes

5. Review Comments to the Author

Reviewer #1: This is an interesting study topic and the paper is well-written. Some small changes and additions can be made to help clarify the paper:

1) In the Abstract, please add that the Bland-Altman analyses indicated an overestimation in MVPA and an underestimation in sedentary time by subjective methods.

2) In the Methods, please state where participants were recruited from.

3) In the Methods, please state the specific cut-offs used for converting accelerometer counts to the different PA intensities (sedentary, light, moderate, vigorous, and MVPA) (line 141-142).

4) Please add units to the x- and y-axis in Figure 2 – e.g., MVPA mpd

5) In the Results, please state the measurement tool (i.e., subjective) that is displaying an overestimation in MVPA (line 255-256) and an underestimation in sedentary time (line 263-264).

6. PLOS authors have the option to publish the peer review history of their article (what does this mean?). If published, this will include your full peer review and any attached files.

Reviewer #1: No

---

## [Author Response · Author response to Decision Letter 0]

28 Aug 2023

Thank you to the reviewer for the comments and feedback. Our response to each suggestion is detailed below. 

1. In the Abstract, please add that the Bland-Altman analyses indicated an overestimation in MVPA and an underestimation in sedentary time by subjective methods.

a. The following sentence was added to the abstract: Bland-Altman analyses indicated that subjects over-reported MVPA and underestimated sedentary time using the IPAQ.

2. In the Methods, please state where participants were recruited from.

a. The following information was added to the methods: Participants were invited to attend two sessions in Brisbane, Queensland, Australia.

3. In the Methods, please state the specific cut-offs used for converting accelerometer counts to the different PA intensities (sedentary, light, moderate, vigorous, and MVPA) (line 141-142).

a. The data have been added to the Methods section. …ActiGraph accelerometers (sedentary < 2860, light = 2861 - 4835, moderate = 4836 – 8452 cpm, vigorous = > 8453).

4. Please add units to the x- and y-axis in Figure 2 – e.g., MVPA mpd

a. ‘mpd’ has been added to axes in Figure 2

5. In the Results, please state the measurement tool (i.e., subjective) that is displaying an overestimation in MVPA (line 255-256) and an underestimation in sedentary time (line 263-264).

a. ‘(IPAQ)’ has been added after the words over-reporting and under-reporting. …though Bland-Altman analysis highlighted the over-reporting (IPAQ) of MVPA by a mean of 17 mpd…; …Bland-Altman analysis highlighted the under-reporting (IPAQ) of sedentary time by a mean of 147 mpd…

---

## [Decision Letter · Decision Letter 1]

25 Sep 2023

Objectively and subjectively measured physical activity levels in individuals with whiplash associated disorder and aged-matched healthy controls

PONE-D-23-11007R1

Dear Dr. Ritchie,

We’re pleased to inform you that your manuscript has been judged scientifically suitable for publication and will be formally accepted for publication once it meets all outstanding technical requirements.

Kind regards,

Emiliano Cè

Academic Editor

PLOS ONE

Additional Editor Comments (optional):

Reviewers' comments:

Reviewer's Responses to Questions

**Comments to the Author**

1. If the authors have adequately addressed your comments raised in a previous round of review and you feel that this manuscript is now acceptable for publication, you may indicate that here to bypass the “Comments to the Author” section, enter your conflict of interest statement in the “Confidential to Editor” section, and submit your "Accept" recommendation.

Reviewer #1: (No Response)

2. Is the manuscript technically sound, and do the data support the conclusions?

Reviewer #1: Yes

3. Has the statistical analysis been performed appropriately and rigorously? 

Reviewer #1: Yes

4. Have the authors made all data underlying the findings in their manuscript fully available?

Reviewer #1: Yes

5. Is the manuscript presented in an intelligible fashion and written in standard English?

Reviewer #1: Yes

6. Review Comments to the Author

Reviewer #1: I advise revising the final line of the conclusion in the abstract (line 51) to be more similar to the final sentence in the conclusion in the manuscript (line 352-354). While accelerometers are generally considered to be more accurate and reliable than subjective methods, we know that accelerometers don't always display 100% accuracy. Therefore, instead of stating "IPAQ may not be a reliable measure of habitual PA in WAD" (line 51) in the abstract, it would be better to conclude that more research is needed to determine the most appropriate methods for assessing PA in WAD.

7. PLOS authors have the option to publish the peer review history of their article (what does this mean?). If published, this will include your full peer review and any attached files.

Reviewer #1: No

---

## [Editor Report · Acceptance letter]

27 Sep 2023

PONE-D-23-11007R1 

Objectively and subjectively measured physical activity levels in individuals with whiplash associated disorder and aged-matched healthy controls. 

Dear Dr. Ritchie:

I'm pleased to inform you that your manuscript has been deemed suitable for publication in PLOS ONE. Congratulations! Your manuscript is now with our production department. 

Kind regards, 

on behalf of

Prof. Emiliano Cè 

Academic Editor

PLOS ONE